# Innovation Research on Symbiotic Relationship of Organization's Tacit Knowledge Transfer Network

Jiang Xu, Huihui Wu 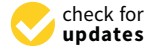 and Jianhua Zhang *

School of Management Engineering, Zhengzhou University, Zhengzhou 450001, China; xujiang@zzu.edu.cn (J.X.); a15139201785@163.com (H.W.)
* Correspondence: tjzhangjianhua@163.com; Tel.:+86-136-8371-2490

**Abstract:** The sustainable development of organizations is inseparable from innovation, and tacit knowledge is the core resource used to achieve organizational innovation. Due to the implicitness of tacit knowledge and the complexity of members' relationships, symbiotic relationships between members have dramatically affected the transfer effect of tacit knowledge. However, previous studies on tacit knowledge transfer only focus on the characteristics of the subject or object; fewer consider the role of symbiotic relationships between knowledge subjects. An organization's tacit knowledge transfer network (OTKTN) is a dynamic knowledge transfer network established among multiple members. Tacit knowledge transfer and sharing among network members conform to the symbiotic feature. To examine various relationships between members, and to investigate the mechanisms that impact tacit knowledge transfer, this article aims to analyze the symbiotic relationships in OTKTN based on the symbiotic perspective. The Lotka–Volterra model was used to construct symbiotic evolution model, and symbiotic coefficients were constructed from the four levels: knowledge-based psychological personal ownership (KPPO) of the knowledge provider, media richness, trust of the knowledge receiver, and organizational rewards matching, to discuss symbiotic modes. Finally, numerical simulation software was applied to simulate the evolution of knowledge levels in members. The results show that the four kinds of symbiotic modes between members include independence, commensalism, asymmetric mutualism, and symmetric mutualism. Symmetric mutualism is the best mode. In this mode, maximum level in independence mode affects the final stable knowledge level; the initial knowledge amount and natural growth rate both affect knowledge growth rate. Media richness, receiver's trust, and organizational rewards matching can increase members' tacit knowledge, but the knowledge provider's KPPO inhibits members' tacit knowledge growth. This article provides guidance to form a healthy symbiotic relationship and help organizations increase tacit knowledge.

**Keywords:** tacit knowledge transfer; symbiotic relationship; symbiotic mode; Lotka–Volterra model; sustainable organizational development

## 1. Introduction

Cowan et al. [1] pointed out that the transfer of explicit knowledge within an organization will lead to homogenization, whereas the transfer of tacit knowledge will lead to a higher growth rate of knowledge. Therefore, tacit knowledge transfer is the primary means of organizational innovation and sustainable development. Tacit knowledge transfer among members refers to sharing part of hidden tacit knowledge to others, who then absorb, apply and internalize their knowledge [2]. Multiple organizational members transfer tacit knowledge through formal or informal channels then form the organization's tacit knowledge transfer network (OTKTN) [3]. From the dynamic perspective, OTKTN is a working system: tacit knowledge in the network flows and transfers [4]; from the perspective of purpose, OTKTN is a function: realizing tacit knowledge sharing and integration, promoting knowledge innovation and value creation [5]. Making an in-depth

analysis of OTKTN can help clarify the flow path of tacit knowledge among members and, more importantly, maximize the utility of existing tacit knowledge [6] and create new knowledge [7].

Previous studies on the factors influencing tacit knowledge transfer in a network mainly focused on analyzing the micro-characteristics of the subject or object, such as individual, media, and organizational characteristics. They paid less attention to the influence of the symbiosis of knowledge subjects. Symbiosis refers to a dependence relationship in which the organisms in the ecosystem exchange resources to adapt to the complex and changeable environment, and move towards unity gradually [8]. Symbiosis is the foundation of constructing an ecosystem and the core of realizing the overall value and system benefit [9,10]. The long-term stable symbiotic relationship within the organization not only promotes the development of the symbiotic subject itself but also enhances its ability to adapt to the uncertainty and disorderly competition of the external environment. The multiple effects of mutual empowerment release more value, and play a crucial role in expanding organizational scale, and promoting organizational sustainability and high-quality development [11]. At present, many scholars have applied symbiosis theory to the fields of economics [12,13] and biology [14,15], which provide a new perspective for the study of tacit knowledge transfer within OTKTN. OTKTN is a dynamic knowledge ecosystem in which members transfer heterogeneous tacit knowledge based on competitive and cooperative relationships [16]. Knowledge subjects take tacit knowledge as an exchange unit. They are interdependent and develop coordination, which conforms to the symbiotic characteristics of the ecosystem [17]. Therefore, this study analyzes the tacit knowledge transfer process in OTKTN from the symbiotic perspective and explores the influence mechanism of symbiotic relationships on the tacit knowledge transfer path. Doing so will be advantageous, helping to guide formation of a stable relationship between members and facilitating sustainable development of the organization.

Based on these objectives, this article establishes the theoretical framework, including the symbiotic system model and the process model of tacit knowledge transfer, constructs the symbiotic evolution model to discuss the symbiotic modes, and uses MATLAB R2016a to simulate the results. Corresponding to these research objectives are the following three research questions addressed in this article:

(1)　What are the specific influence factors in the tacit knowledge transfer process?
(2)　What are the symbiotic modes among members? Which one is the best mode?
(3)　What are the individual-related factors influencing tacit knowledge growth in the best mode?

The remainder of this study is arranged as follows. Section 2 is the literature review, introducing the factors affecting tacit knowledge transfer and the research method of OTKTN. Section 3 is the theoretical framework; it describes the elements of the symbiotic system, and builds the conceptual model and the process model of tacit knowledge transfer of OTKTN. Section 4 contains the methodology that explains the construction of the symbiotic model and the classification of symbiotic modes. Section 5 presents the numerical simulation and results. Section 6 provides research conclusions, the theoretical and practical significance, limitations, and prospects.

## 2. Literature Review

### 2.1. The Influencing Factors of Tacit Knowledge Transfer

Previous research has expounded the factors influencing of tacit knowledge transfer in networks from three aspects, namely, individual characteristics [18–25], media characteristics [26–28], and organizational characteristics [29–31].

Some researchers discussed the factors influencing individual characteristics (including both knowledge providers and knowledge receivers). In terms of knowledge providers: Wu et al. [18] conducted a study on 32 virtual educational communities and found that knowledge-based psychological ownership negatively affects providers' behavior of tacit knowledge transfer. Peng [19] found that knowledge-based psychological ownership

positively impacts providers' knowledge hiding behavior. Bhattacharya [20] surveyed 429 employees from four basic industries: audio-visual, mechanical manufacturing, pharmaceuticals, and telecommunications. The study also found that knowledge-based psychological ownership increased providers' tacit knowledge hiding behavior and inhibited the growth of tacit knowledge. The study by Tian [21] has reached consistent conclusions. In terms of knowledge receivers, Wang et al. [22] believed that knowledge receivers tend to seek help from people with an emotional trust foundation so that it is easier to understand, master, and absorb the tacit knowledge. Holste and Fields [23] explored the relationship between cognitive trust and the willingness to use tacit knowledge among professionals. The result showed that cognitive trust could promote the willingness to use tacit knowledge. Knowledge receivers are more willing to accept and use knowledge when they believe that the provider is capable. Alexopoulos and Buckley [24] divided receivers' trust into professional and personal trust, and proposed that the two kinds of trust both promote tacit knowledge transfer effectively. Santoro and Saparito [25] explored the knowledge transfer between university and industry. It stated that relational trust is positively associated with knowledge transfer; as knowledge becomes more tacit, the positive effect of trust becomes stronger.

Some researchers have explored the influencing factors at the media level. For example, the study by Albino et al. [26] stressed that knowledge transfer media could reduce the fuzziness and uncertainty of knowledge and increase knowledge transfer effectiveness. Daft et al. [27] conducted field research of middle-level and senior managers to explain managers' choice of media. It showed that managers use rich media for fuzzy communication. High-richness media are more conducive to reducing the ambiguity of information. Daft and Lengel [28] illustrated that high-richness media could provide quick feedback, spread a variety of hints, and convey personal feelings. Therefore high-richness media is more conducive to promoting organizational learning and reducing information ambiguity than low-richness media; low-richness media is more conducive to processing easily understood information and standardized data.

Studies have also been conducted to show the influencing factors of the organization. Super et al. [29] studied the role of compensation based on group performance in stimulating cognitive motivation and pro-social motivation by using the motivation information processing (MIP-G) model in groups. The results show that material reward can promote tacit knowledge transfer among team members by strengthening the pro-social motivation of members. Gagne [30] believed that both material awards and spiritual incentives could promote knowledge transfer among individuals within the organization. Husted and Michailova [31] stressed that organizations could alleviate or overcome hostility in knowledge transfer by encouraging and exciting knowledge sharing among employees. The study also highlighted that the primary strategy for enterprises to transfer knowledge is adjustment of incentive mechanisms and development of a knowledge-sharing culture.

Through reviewing prior studies about tacit knowledge transfer in networks, it can be seen that they only focus on the impact of the characteristics of the subject or object; there is a gap in exploring the symbiotic relationships between knowledge subjects. It is well known that tacit knowledge transfer is a knowledge-sharing behavior. In the process of knowledge transfer, various relationships between members are involved. They are interdependent and mutually restricting. Elfring et al. [32] considered that inter-social network relations between members would affect the effectiveness of knowledge transfer. Reagan et al. [33] also pointed out that the strong connection between knowledge subjects is conducive to transferring tacit knowledge. Therefore, it is necessary to conduct a deeper study of members' relationships to fill the gap. OTKTN is built based on the symbiosis of members. Thus, this article clarifies how symbiotic relationships impact tacit knowledge transfer in OTKTN. In addition, this article explores the tacit knowledge transfer process based on conclusions of previous studies to construct symbiotic coefficients, which lay a good foundation for the discussion of symbiotic modes.

*2.2. Organization's Tacit Knowledge Transfer Network Research Method*

Questionnaires and empirical research are most commonly used methods in previous studies on the discipline of tacit knowledge transfer. For example, Li and Zhu [34], Li and Hsieh [35], Wang et al. [36], and Cummings and Teng [37] used the questionnaire method to analyze the influencing factors of tacit knowledge transfer. Ning and Fang [38] adopted empirical research to analyze the tacit knowledge transfer model in knowledge alliances and proposed concrete strategies to promote tacit knowledge transfer. Rotsios et al. [39] conducted an empirical study on the Greek international joint ventures (I.J.V.) operated in southeast Europe to investigate the influence of trust on knowledge transfer and expected benefits from knowledge transfer.

In addition, some scholars even use a quantitative model to analyze tacit knowledge transfer. For example, Huang and Yang [40] constructed a safety knowledge transfer model to explore the relationship between safety knowledge transfer and safe working environments. Zhao et al. [41] explained the influence of four dimensions of perceived value on tacit knowledge transfer willingness by using the improved T.A.M. (Technology Acceptance Model).

However, the above methods or models have certain limitations in sorting out the influence of symbiosis on the evolution of tacit knowledge amount. The Lotka–Volterra model (L–V) can provide a scientific tool. The Lotka–Volterra model was proposed by Afred Lotka of the United States and Vito Volterra of Italy [42]. This model was initially used to study predator–prey relationships in the ecosystem. Due to the lack of consideration of practical factors, it is difficult to simulate the actual ecological environment. Later, scholars combined the L–V model with the logistic model. At present, the L–V model is mainly used to elucidate the interaction between different subjects and their dynamic relationships with the environment in complex systems [43]. For example, the L–V model has been used to investigate the symbiotic relationships between innovation ecosystems [44–47] and economies [48,49]. However, the L–V model was seldom applied in tacit knowledge transfer. Srivastava et al. [16] indicated that the tacit knowledge transfer network is linked by the flow and integration of tacit knowledge resources; members in the network are interdependent and co-evolutionary, which shows the symbiotic characteristics of biology. Therefore, the L–V model can be applied to analyze the symbiotic relationships in OTKTN.

The innovations of this article include: (1) This article explores the influence mechanism of symbiosis on tacit knowledge transfer based on the symbiotic perspective. (2) Through detailed analysis of the knowledge transfer process, the symbiotic coefficients are constructed from four levels, including providers, receivers, media, and organization, to explain the symbiotic effects among members. (3) Numerical simulation is used to simulate the tacit knowledge transfer process in order to solve the problem that research data are challenging to obtain.

## 3. Theoretical Framework

*3.1. Conceptual Model of Organization's Tacit Knowledge Transfer Network*

Symbiosis theory emphasizes that symbiosis refers to the interdependence and joint development of two or more closely connected species or populations and environment; populations with complementary resources form a cooperative symbiosis, and thus enable the populations to undergo long-term development; populations competing for the same resources form a competitive symbiosis, thereby expanding or weakening populations [50]. Moreover, the symbiotic system includes three elements: symbiotic unit, symbiotic interface, and symbiotic environment [35].

OTKTN is a unique ecosystem. In the organizational environment, the internal members use tacit knowledge as the exchange unit to search, absorb or transfer knowledge, and gradually develop into differentiated symbiotic modes, showing the symbiotic characteristics. Specifically, the symbiotic unit is the subject of primary energy production and exchange, and the competitive cooperation and evolution of the symbiotic unit are the basis for the formation of a symbiotic system [50]. For OTKTN, the symbiotic unit refers

to the knowledge subjects for tacit knowledge exchange, including knowledge providers and receivers. Providers are knowledge holders. They have the right to decide whether to share knowledge because of the implicit characteristic of tacit knowledge. Their knowledge transfer behaviors are based on emotion or exchange—to obtain more critical tacit knowledge. Receivers are recipients of tacit knowledge. They usually obtain and absorb knowledge by learning providers' shared knowledge. However, in facing the knowledge given by providers, receivers do not fully accept knowledge. A member within OTKTN can be either a knowledge provider or a receiver. Secondly, the symbiotic interface is the connector or information channel of the symbiotic unit and is the carrier of information exchange [50]. In OTKTN, the symbiotic interface is a channel, tool, or media for members to exchange tacit knowledge. Tacit knowledge is not as visible as explicit knowledge, and its media needs to be more abundant to reduce ambiguity. Finally, symbiotic environment refers to factors outside the symbiotic unit. It is the external condition for the synergistic development of symbiotic units [50]. In OTKTN, the symbiotic environment refers to the organization where members transfer tacit knowledge.

In summary, the symbiotic system of OTKTN is shown in Figure 1.

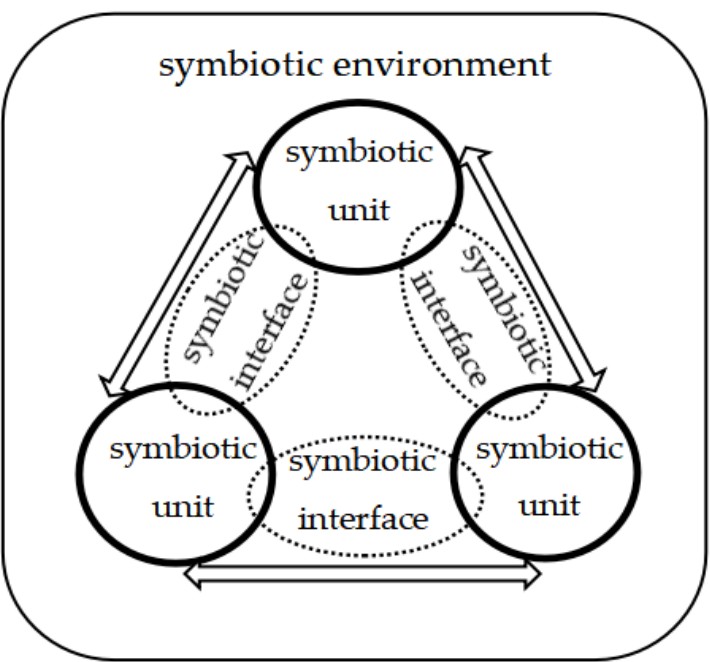

**Figure 1.** The conceptual model of symbiotic system of the organization's tacit knowledge transfer network.

*3.2. The Process Model of Tacit Knowledge Transfer in an Organization's Tacit Knowledge Transfer Network*

The organization's tacit knowledge transfer network is an organizational system of tacit resource exchange and sharing. The coordinated development of symbiotic subjects is conducive to increasing hidden resources of themselves and organizations. Based on previous research conclusions, this article constructs a process model of tacit knowledge transfer in the network from the symbiotic perspective, as shown in Figure 2. The tacit knowledge transfer process includes knowledge providers, receivers, media, and tacit knowledge. Under the synergy of the organizational environment, members continue to exchange knowledge to maximize their own interests.

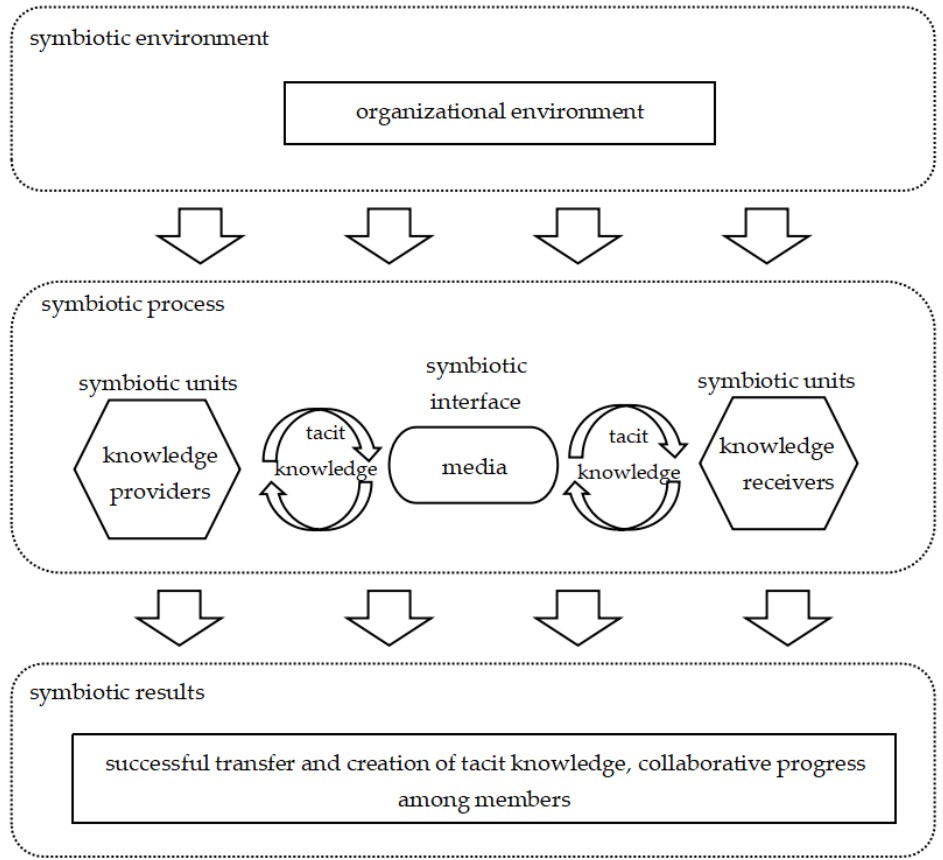

**Figure 2.** The process model of tacit knowledge transfer in the symbiotic network.

## 4. Methodology

This article constructs the symbiotic evolution model of knowledge subjects based on the Lotka–Volterra model, and builds the symbiotic coefficients by analyzing the process of network tacit knowledge transfer to discuss various symbiotic modes.

### 4.1. Model Hypothesis

The Lotka–Volterra model (L–V) model stressed that the population growth law conforms to the Logistic model (S-type growth): The initial growth rate is fast, and the growth rate slows down to a certain extent under specific resource and environmental constraints and tends to be stable until the growth stops reaching the final stable level [51].

In OTKTN, tacit knowledge transfer of members promotes the increase of knowledge amount. However, due to the constraints of resources, technical, and personal ability, the growth rate of knowledge slows down until it reaches saturation. Therefore, the amount of members' tacit knowledge conforms to the Logistic growth law. In this way, this article uses the L–V model to explore the evolution of tacit knowledge amount in OTKTN. This article presents the following hypotheses before modeling:

**Hypothesis 1 (H1).** *OTKTN comprises members 1 and 2, and the two members are independent and interdependent. In independence mode, set the natural growth rate as r, which indicates the growth rate of tacit knowledge; set the amount of tacit knowledge as n (t), which symbolizes that knowledge amount changes with time t; set the maximum level as N, which indicates the maximum value, and the growth rate of knowledge gradually slows down and tends to 0 in this time.*

**Hypothesis 2 (H2).** *There are interactions among members in OTKTN, which are expressed by the symbiotic coefficient. Because of the viscosity of tacit knowledge, a member's behavior will increase or have no impact on others' tacit knowledge amount, but not lead to a knowledge reduction. So,*

*the symbiotic coefficient is only greater than 0 or equal to 0. If a member's symbiotic coefficient is greater than 0, others positively impact it; if the symbiotic coefficient is 0, others have no impact on it.*

**Hypothesis 3 (H3).** *When members' marginal output equals marginal income, knowledge amount stops growing and reaches the maximum level.*

### 4.2. Model Construction

Set $n_1(t)$ and $n_2(t)$ be the tacit knowledge amount of members 1 and 2 at time t; set $r_1$ and $r_2$ as natural growth rates for members 1 and 2, $r_1 > 0$, $r_2 > 0$; set $N_1$ and $N_2$ as the maximum level for members 1 and 2 in independence.

#### 4.2.1. Construction of Two Subject Evolution Model in Independence Phase

Members learn knowledge independently and do not rely on other members. The evolutionary dynamic equations of tacit knowledge amount of members are:

$$\begin{cases} \frac{dn_1}{dt} = r_1 n_1 \left(1 - \frac{n_1}{N_1}\right), \; n_1(0) = n_{10} \\ \frac{dn_2}{dt} = r_2 n_2 \left(1 - \frac{n_2}{N_2}\right), \; n_2(0) = n_{20} \end{cases} \tag{1}$$

Among them, $r_1 n_1$ and $r_2 n_2$ represent the growing trend of members' knowledge amount; $1 - \frac{n_1}{N_1}$ and $1 - \frac{n_2}{N_2}$ represent the Logistic coefficient of members, that is, the blocking effect of resource constraints and organizational environment; $n_{10}$ and $n_{20}$ represent the initial knowledge amount of members 1 and 2.

#### 4.2.2. Construction of a Two Subject Evolution Model in Symbiosis Phase

The organization's incentive system and hierarchical structure reinforce the competition of internal members, and the increase in remuneration and advancement of members depends on their performance vis-à-vis other members [52]. Tacit knowledge guarantees individual value and status; more importantly, it is the source of transcending colleagues and obtaining promotions [53]. Under competitive pressure, sharing tacit knowledge may result in lots of similar knowledge occurring in a short time [54]. Individuals will lose their unique competitive advantage. To avoid this phenomenon, individuals usually hide knowledge [55]. Bunderson et al. [56] emphasized that competition may encourage employees to strive to improve their ability and performance and hide their knowledge to surpass their opponents in competition and obtain a higher status. However, OTKTN is a multi-member coexistence system. To obtain more key tacit knowledge for long-term development, members must cooperate with relevant individuals to occupy a favorable position [57]. So members will choose a part of tacit knowledge to share. The remaining part belongs to themselves, which becomes "private property". At this time, members' knowledge-based psychological personal ownership (KPPO) is generated. Bhattacharya [20] stressed that KPPO increased providers' tacit knowledge hiding behavior. Wu et al. [18] believed that KPPO negatively affects providers' tacit knowledge transfer willingness. Thus, it can be seen that a provider's knowledge contribution is related to its KPPO. If the KPPO degree of provider is expressed as $\alpha$, $0 \leq \alpha \leq 1$, the knowledge contribution rate is $1 - \alpha$. When $\alpha = 1$, the provider's KPPO is the strongest, privatizing knowledge without sharing, the knowledge contribution degree is 0. When $\alpha = 0$, the provider believes knowledge does not belong to the individual and actively shares knowledge, the contribution degree is 1.

Szulanski [58] stressed that effective knowledge transfer should ideally mean that all knowledge is retained, but transfer obstacles mean that not all knowledge can always be transferred. The reason is the lack of media or improper media selection. For example, Ounjian [59] noted that communication media affects knowledge transfer, rich media can reduce knowledge loss. Simonin [60] pointed out that the use of information technology increases knowledge transfer and acceptance amount. Unlike explicit knowledge, tacit

knowledge (such as valuable experience and techniques) cannot be transmitted and pre-served through concrete carriers such as words and pictures. With the increasing degree of tacit degree, knowledge becomes more vague and difficult to articulate, which increases the transfer cost and loss rate [61,62], and needs richer media to transfer [63]. High-richness media is more conducive to reducing ambiguity [27] and improving reduction, making knowledge transfer convenient and efficient. In this way, it can be seen that media richness affects knowledge reduction in the process of tacit knowledge transfer. If the degree of media richness is expressed by $\beta$, $0 \leq \beta \leq 1$, it indicates the influence of media on knowledge reduction rate; the ratio of knowledge transferred by the provider through media is: $(1 - \alpha) \times \beta$.

Szulanski and Cappetta [64] found that when the knowledge providers are considered unreliable or unknowledgeable by receivers, their suggestions and demonstrations are likely to be challenged and resisted, and it is hard to transfer knowledge successfully. Conversely, when receivers feel the providers are trustworthy, they will be more willing to accept information [65]. Receivers' trust makes knowledge transfer activities easier [66]. It is able to effectively increase the tacit knowledge amount that providers transfer and receivers absorb [67]. Thus, it is clear that a receiver's knowledge absorption is affected by its trust in the provider. If the receiver's trust degree is expressed as $\gamma$, $0 \leq \gamma \leq 1$, it indicates the influence of receiver's trust on knowledge absorption rate, then the ratio of knowledge transferred is: $(1 - \alpha) \times \beta \times \gamma$.

Set $\varphi$ as the symbiotic coefficient of member 1 and member 2, reflecting the interaction effect of members 1 and 2. The effect of member 2 on member 1 can be expressed as $\varphi_{12} = (1 - \alpha_2) \times \beta \times \gamma_1$; the effect of member 1 on member 2 can be expressed as $\varphi_{21} = (1 - \alpha_1) \times \beta \times \gamma_2$. In this way, when OTKTN generates tacit knowledge transfer, the evolutionary dynamic equations of tacit knowledge amount are:

$$\begin{cases} \frac{dn_1}{dt} = r_1 n_1 \left(1 - \frac{n_1}{N_1} + \varphi_{12} \frac{n_2}{N_2}\right), \ n_1(0) = n_{10} \\ \frac{dn_2}{dt} = r_2 n_2 \left(1 - \frac{n_2}{N_2} + \varphi_{21} \frac{n_1}{N_1}\right), \ n_2(0) = n_{20} \end{cases} \tag{2}$$

Among them, $\varphi_{12}$ represents the influence of member 2 on member 1, and $\varphi_{21}$ represents the influence of member 1 on member 2.

### 4.2.3. Construction of Two Subject Evolution Model When Considering Organizational Rewards

In OTKTN, tacit knowledge transfer is a knowledge interaction behavior based on competition and cooperation symbiosis. The ideal state is that members share knowledge without reservation [68]. However, when providers share tacit knowledge, they need to make explicit coding first then share it [69]. In the process of explicit coding, knowledge providers should not only cost time and energy, but also inject subjective thinking and emotion [70]. So, Szulanski [58] found that knowledge providers are unwilling to share tacit knowledge with others, because: (1) they are afraid to lose ownership and dominant position of crucial knowledge, (2) the return from transferring knowledge is very limited, and (3) support for knowledge transfer is insufficient, and they are unwilling to invest a lot of time and resources. Obviously, competition and cooperation as internal driving forces are not enough to optimize the symbiotic evolution of tacit knowledge among members. Gagne [30] believed that material awards and spiritual incentives could promote knowledge transfer among individuals. Zhao et al. [71] also suggested that for inactive members unwilling to share tacit knowledge actively, organizational rewards increase their desire to help others and promote knowledge transfer. Therefore, organizational reward as an external incentive outside members' relationships plays a vital role in stimulating members' tacit knowledge transfer willingness. The reason is that organizational rewards convey the support and encouragement of organizations for knowledge transfer. This encouragement signal will give providers more pleasure and increase sharing willingness [72].

Expectancy theory emphasizes that people's motivation to take action depends on their value assessment of the outcome (rewards) and the matching degree between outcome and individual needs [73]. Therefore, more attention should be paid to the matching degree of rewards and individual needs in addition to considering organizational rewards. ERG theory divides personal needs into three categories: survival, relationship, and development needs [74]. Organizational rewards include external incentives (e.g., salary and bonus) and internal incentives (e.g., training, further education, and ability improvement). In terms of the high level of demand, the greater the incentive depth, the better the incentive effect maintains [74]. Thus, the matching degree of organizational rewards is able to promote the knowledge transfer of providers. If the influence of matching degree of organizational rewards on knowledge providers is expressed as $\eta$, $\eta \geq 0$, then the ratio of tacit knowledge transferred under the influence of organizational rewards is $\delta = (1 - \alpha) \times \beta \times \gamma + \eta$. Thus, when there is an organizational reward, the evolutionary dynamics equations of member tacit knowledge amount are:

$$\begin{cases} \frac{dn_1}{dt} = r_1 n_1 \left(1 - \frac{n_1}{N_1} + \delta_{12} \frac{n_2}{N_2}\right), \ n_1(0) = n_{10} \\ \frac{dn_2}{dt} = r_2 n_2 \left(1 - \frac{n_2}{N_2} + \delta_{21} \frac{n_1}{N_1}\right), \ n_2(0) = n_{20} \end{cases} \quad (3)$$

Among them, $\delta_{12}$ represents the influence of member 2 on member 1, and $\delta_{21}$ represents the influence of member 1 on member 2.

### 4.2.4. Summary

This article investigated the process of tacit knowledge transfer in the network in detail. It is concluded that the symbiotic relationship of members is affected by the provider's knowledge-based psychological personal ownership (KPPO), media richness, receiver's trust, and organizational rewards matching. The symbiotic coefficient is the combined effects of these factors, that of members 1 and 2 can be expressed as $\delta_{12} = (1 - \alpha_2) \times \beta \times \gamma_1 + \eta_2$, $\delta_{21} = (1 - \alpha_1) \times \beta \times \gamma_2 + \eta_1$. $\delta_{12} = (1 - \alpha_2) \times \beta \times \gamma_1 + \eta_2$ represents the influence of member 2's knowledge transfer activities on member 1's absorbed knowledge amount when member 2 is a knowledge provider, and member 1 is a receiver. $\alpha_2$ is the coefficient of KPPO of member 2, and $1 - \alpha_2$ is the knowledge contribution degree of member 2 impacted by KPPO; $\beta$ is the media richness coefficient, symbolizing the knowledge reduction degree impacted by the media; $\gamma_1$ is the degree of member 1's trust in member 2, meaning the knowledge absorption degree impacted by the trust of member 1; $\eta_2$ is the matching degree of organizational reward, which indicates the knowledge transfer rate of member 2 increased by organizational reward. $\delta_{21} = (1 - \alpha_1) \times \beta \times \gamma_2 + \eta_1$, represents the influence of member 1's knowledge transfer on member 2's absorption when member 1 is a knowledge provider, and member 2 is a receiver. The analysis process is the same as above.

### 4.2.5. Construction of Multi-Member Evolution Model

When considering the symbiotic relationships between member A and multiple members, it is assumed that X members are cooperating with member A, the influence of member i on member A is $\delta_{Ai}$, the influence of member A on member i is $\delta_{iA}$, and the evolutionary dynamic equations of tacit knowledge amount symbiotic with multiple members are:

$$\begin{cases} \frac{dn_A}{dt} = r_A n_A \left(1 - \frac{n_A}{N_A} + \sum_{i=1}^{X} \delta_{Ai} \frac{n_i}{N_i}\right), \quad n_A(0) = n_{A0} \\ \frac{dn_1}{dt} = r_1 n_1 \left(1 - \frac{n_1}{N_1} + \delta_{1A} \frac{n_A}{N_A} + \sum_{i=2}^{X} \delta_{1i} \frac{n_i}{N_i}\right), \quad n_1(0) = n_{10} \\ \frac{dn_2}{dt} = r_2 n_2 \left(1 - \frac{n_2}{N_2} + \delta_{2A} \frac{n_A}{N_A} + \delta_{21} \frac{n_1}{N_1} + \sum_{i=3}^{X} \delta_{2i} \frac{n_i}{N_i}\right), \quad n_2(0) = n_{20} \\ \quad\quad\quad\quad \cdots\cdots \\ \frac{dn_X}{dt} = r_X n_X \left(1 - \frac{n_X}{N_X} + \delta_{XA} \frac{n_A}{N_A} + \sum_{i=1}^{X-1} \delta_{Xi} \frac{n_i}{N_i}\right), \quad n_X(0) = n_{X0} \end{cases} \quad i = 1, 2, 3, \ldots, X \quad (4)$$

### 4.3. Stability Analysis of the Model and Symbiotic Modes Discussion

Members within the organization freely share knowledge and eventually evolve to a dynamic equilibrium state. Set $\frac{dn_1}{dt} = 0$, $\frac{dn_2}{dt} = 0$, the stability of the equilibrium point is analyzed, and the local stability points are obtained as follows:

$$\text{P1 }(0, 0),\ \text{P2 }(N_1, 0),\ \text{P3 }(0, N_2),\ \text{P4 }\left(\frac{N_1(1+\delta_{12})}{1-\delta_{12}\delta_{21}}, \frac{N_2(1+\delta_{21})}{1-\delta_{12}\delta_{21}}\right) \tag{5}$$

The Jacobian matrix of the evolutionary system is:

$$J = \begin{bmatrix} r_1\left(1 + \delta_{12}\frac{n_2}{N_2} - 2\frac{n_1}{N_1}\right) & r_1\delta_{12}\frac{n_1}{N_2} \\ r_2\delta_{21}\frac{n_2}{N_1} & r_2\left(1 + \delta_{21}\frac{n_1}{N_1} - 2\frac{n_2}{N_2}\right) \end{bmatrix} \tag{6}$$

When the equilibrium points make the determinant Det (J) > 0 and trace Tr (J) < 0 of the Jacobian matrix, it is a stable equilibrium point. The stability conditions are shown in Table 1.

**Table 1.** Symbiotic evolution equilibrium points and stability conditions.

| Equilibrium Points | Det (J) | Tr (J) | Stability Conditions |
|---|---|---|---|
| $P_1\ (0, 0)$ | $r_1r_2$ | $r_1+r_2$ | Unstable |
| $P_2\ (N_1, 0)$ | $-r_1r_2(1+\delta_{21})$ | $-r_1+r_2(1+\delta_{21})$ | $\delta_{21} < -1$ |
| $P_3\ (0, N_2)$ | $-r_1r_2(1+\delta_{12})$ | $-r_2+r_1(1+\delta_{12})$ | $\delta_{12} < -1$ |
| $P_4\left(\frac{N_1(1+\delta_{12})}{1-\delta_{12}\delta_{21}}, \frac{N_2(1+\delta_{21})}{1-\delta_{12}\delta_{21}}\right)$ | $\frac{r_1r_2(1+\delta_{12})(1+\delta_{21})}{1-\delta_{12}\delta_{21}}$ | $\frac{-r_1(1+\delta_{12})-r_2(1+\delta_{21})}{1-\delta_{12}\delta_{21}}$ | $\delta_{12} < 1$<br>$\delta_{21} < 1$ |

$\delta_{12}$ and $\delta_{21}$ are interdependence coefficients between two members, namely, the influence of symbiotic relationship of the two members on their evolution. The value ranges of $\delta_{12}$ and $\delta_{21}$ decide symbiotic modes between the two members as seen in Table 2. It can be seen that the equilibrium points of commensalism mode, asymmetric mutualism mode, and symmetric mutualism mode are all the point P4 $\left(\frac{N_1(1+\delta_{12})}{1-\delta_{12}\delta_{21}}, \frac{N_2(1+\delta_{21})}{1-\delta_{12}\delta_{21}}\right)$, and the equilibrium points of symbiotic evolution of members' knowledge amount are affected by the symbiotic coefficient and maximum level between members.

**Table 2.** Member symbiotic evolution modes.

| Value Combination | Symbiotic Mode | Explanation |
|---|---|---|
| $\delta_{12} = \delta_{21} = 0$ | Independence | Members have no impact on each other. Their own resources and conditions determine members' knowledge growth. Knowledge does not flow in the organization. |
| $\delta_{12} = 0,\ 0 < \delta_{21} < 1$ or $\delta_{21} = 0,\ 0 < \delta_{12} < 1$ | Commensalism | Members with positive symbiotic coefficient gain, the members with 0 have no change, and the network has no compensation mechanism for non-profit parties. |
| $0 < \delta_{12}, \delta_{21} < 1$ $\delta_{12} \neq \delta_{21}$ | Asymmetric mutualism | There is a wide range of gain in the symbiotic network, and mutual promotion among members, multilateral flow of knowledge, and knowledge resources generally increase, but the symbiotic coefficients lead to different growth rates. |
| $0 < \delta_{12}, \delta_{21} < 1$ $\delta_{12} = \delta_{21}$ | Symmetric mutualism | There is a wide range of gain in the symbiotic network, multilateral flow of knowledge, equal increase of members' knowledge resources, and synchronization of members' knowledge growth. |

### 5. Model Simulation and Results

Different value combinations of symbiotic coefficients produce different symbiotic modes. The numerical simulation by MATLAB R2016a can directly describe the evo-

lution path of tacit knowledge amount under various modes. Set the natural growth rates of members 1 and 2 as $r_1 = r_2 = 1$; set the initial knowledge amount of members 1 and 2 as $n_{10} = n_{20} = 50$; set the maximum level of knowledge in independence as $N_1 = N_2 = 800$ and the evolution period is assigned as t = 30. The following sections discuss the influence of members' symbiotic modes and correlative factors on tacit knowledge amount in knowledge transfer.

### 5.1. Commensalism Mode

Taking $\delta_{12} = 0.3$, $\delta_{21} = 0$. The symbiotic mode is commensalism. The evolution results are shown in Figure 3. It can be seen from Figure 3 that the final stable knowledge level of member 1 is greater than the maximum level in independence (i.e., $N_1 = 800$); the final stable knowledge level of member 2 is equal to the maximum level in independence (i.e., $N_2 = 800$). The difference between them is the symbiotic coefficient, that of member 1 is greater than 0 (i.e., $\delta_{12} = 0.3$), and that of member 2 is equal to 0 (i.e., $\delta_{21} = 0$), indicating that member 2 has a positive impact on member 1, while member 1 has no impact on member 2. Therefore, member 1 benefits from member 2, and its final stable knowledge level is higher than the maximum level, while the final stable knowledge level of member 2 is equal to the maximum level. It can be seen that in commensalism mode, the party with a positive symbiotic coefficient obtains new knowledge, and the party with a symbiotic coefficient of 0 neither obtains additional knowledge nor loses original knowledge.

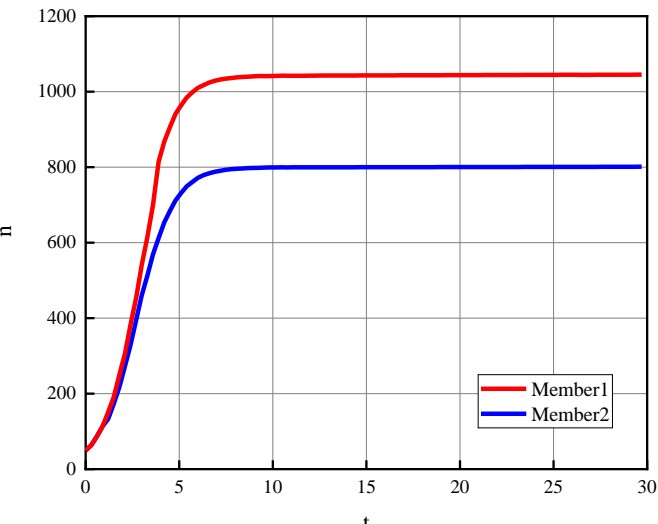

**Figure 3.** The dynamic evolution with commensalism mode.

According to the commensalism mode reflected in Figure 3, there may be adverse effects after a long time. The underlying reasons are as follows: (1) Member 2 has shared knowledge to member 1, but member 1 has no feedback, causing no return for member 2's effort, which is not conducive to developing a stable relationship among members. (2) Tacit knowledge is a kind of unique advantage. When providers transfer knowledge, they not only need take time and invest emotion, but also face the risk of losing unique competitive advantage. However, when member 2 transfers knowledge, there is no return. To avoid this phenomenon and keep the unique advantage, member 2 will reduce knowledge transfer behavior which does not benefit the organizational innovation.

### 5.2. Asymmetric Mutualism Mode

Taking $\delta_{12} = 0.3$, $\delta_{21} = 0.2$. The symbiotic mode is asymmetric mutualism. The evolution results are shown in Figure 4. It can be seen from Figure 4 that when the symbiotic coefficients of members 1 and 2 are greater than 0, the final stable knowledge levels of them both are higher than the maximum levels in their independence

(i.e., $N_1 = N_2 = 800$). Moreover, the final stable knowledge level of 1 is larger than that of 2. The difference between them is the symbiotic coefficient, the coefficients of members 1 and 2 are greater than 0, but that of member 1 is greater than that of member 2 (i.e., $\delta_{12} = 0.3$, $\delta_{21} = 0.2$), indicating that they both have a positive impact on each other, but the impact of member 2 is higher than that of member 1. Therefore, the knowledge increment of member 1 is higher than that of member 2. It reflects that in asymmetric mutualism mode, different symbiotic coefficients lead to different final stable knowledge levels. The party with a larger symbiotic coefficient has a higher knowledge level than that of the smaller one.

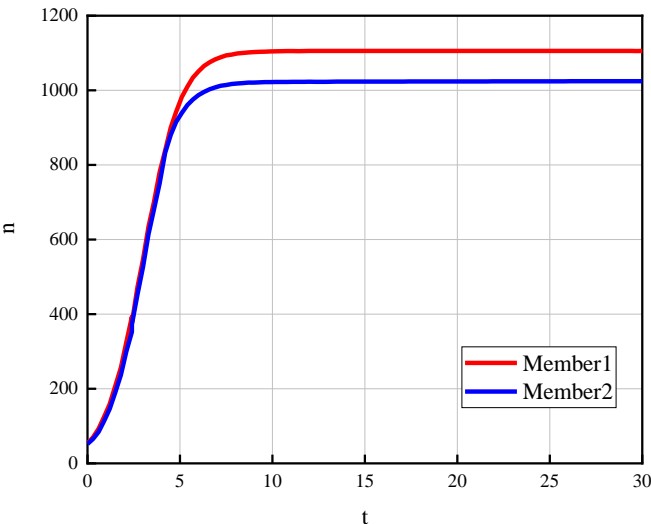

**Figure 4.** The dynamic evolution with asymmetric mutualism mode.

The asymmetric mutualism mode can promote organizational development in the short term, but it is not the optimal mode in the long run. The reason is that members 1 and 2 both benefit, but the symbiotic coefficient of member 1 is greater than that of member 2. This leads to member 1 gaining more and member 2 paying more. In the long run, this payment phenomenon is not proportional to the return, and will reduce member 2's tacit knowledge transfer willingness, and thus is not conducive to increasing tacit knowledge in the network and the sustainable development of the organization.

*5.3. Symmetric Mutualism Mode*

When the symbiotic coefficients of members 1 and 2 are positive and equal ($\delta_{12} = \delta_{21} > 0$), they are in the symmetric mutualism mode. In this mode, the final stable knowledge levels of members 1 and 2 are equal, the payments of the two are both proportional to the return. The organizational tacit knowledge amount increases by transferring between members 1 and 2. Therefore, the symmetric mutualism mode is the best mode for members. Taking $\delta_{12} = \delta_{21} = 0.3$. The following analysis is made on the influence factors of the evolution of tacit knowledge in this mode: the maximum level in independence mode, the natural growth rate, and the initial knowledge amount.

5.3.1. Impact of Maximum Scale

Taking $N_1 = 600$, $N_2 = 800$; $r_1 = r_2 = 1$; $n_{10} = n_{20} = 50$, the evolution results are shown in Figure 5. It can be seen from Figure 5 that the final stable knowledge levels of members 1 and 2 are greater than the maximum levels in their independence (i.e., $N_1 = 600$, $N_2 = 800$), but the final stable knowledge level of member 2 is greater than that of 1. The difference between them is the maximum level. That of member 2 is larger than that of member 1. This shows that the maximum level influences the final stable knowledge level; the larger the maximum level, the bigger it is. Therefore, when other conditions remain unchanged, the more the maximum levels in independence mode, the more knowledge acquired when knowledge transfer.

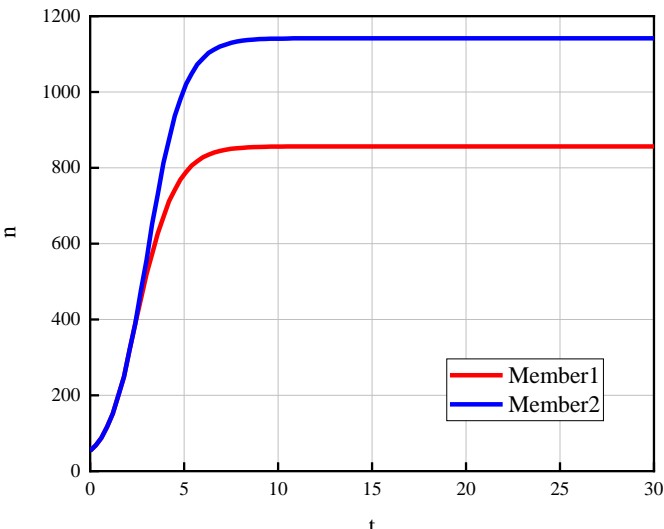

**Figure 5.** Symmetric mutualism mode (different maximum levels).

### 5.3.2. Impact of Natural Growth Rate

Taking $N_1 = N_2 = 800$; $r_1 = 1$, $r_2 = 0.7$; $n_{10} = n_{20} = 50$, the evolution results are shown in Figure 6. It can be seen from Figure 6 that the final stable knowledge levels of members 1 and 2 are equal, and both greater than the maximum levels in their independence (i.e., $N_1 = N_2 = 800$). Whereas the knowledge growth rate of member 1 is faster than that of member 2 before reaching a stable state. The difference between them is the natural growth rate, that of member 1 (i.e., $r_1 = 1$) is greater than that of member 2 (i.e., $r_2 = 0.7$), so the knowledge growth rate of member 1 is faster than that of member 2. This indicates that the natural growth rate only affects the knowledge growth rate and does not affect members' final stable knowledge levels. Therefore, when other conditions remain unchanged, the bigger the knowledge growth rate in independence mode, and the faster the knowledge absorption rate in the knowledge transfer process, but the final stable knowledge level depends on the maximum level in independence mode.

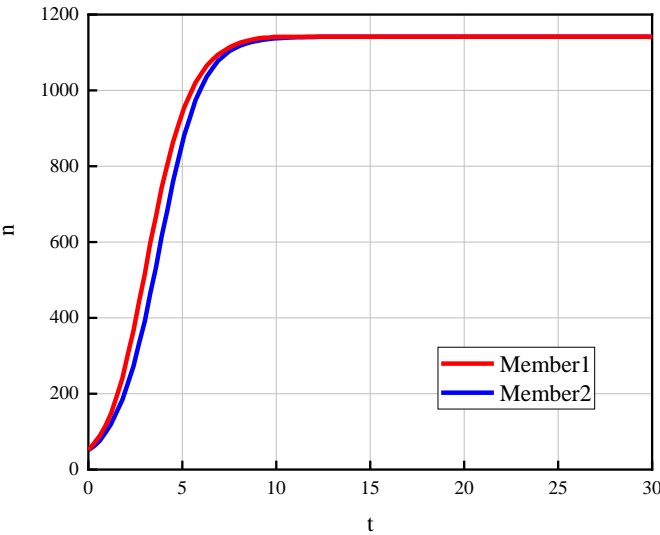

**Figure 6.** Symmetric mutualism mode (different natural growth rates).

### 5.3.3. Impact of Initial Knowledge Amount

Taking $N_1 = N_2 = 800$; $r_1 = r_2 = 1$; $n_{10} = 50$, $n_{20} = 200$, the evolution results are shown in Figure 7. It can be seen from Figure 7 that the final stable knowledge levels of members 1 and 2 are equal, and both greater than the maximum levels in their independence

(i.e., $N_1 = N_2 = 800$). Whereas the knowledge growth rate of member 2 is faster than that of member 1 before reaching a stable state. The difference between the two is the initial knowledge amount. That of member 2 is larger than that of member 1 (i.e., $n_{10} = 50$, $n_{20} = 200$), so the knowledge growth rate of member 2 is faster than that of member 1. It shows that the initial knowledge amount only affects the knowledge growth rate, but the final stable knowledge level has no effect. Therefore, when other conditions remain unchanged, members with a larger initial knowledge amount will understand knowledge more easily, and the knowledge growth rate is fast. However, the final stable knowledge level depends on the maximum level in independence mode.

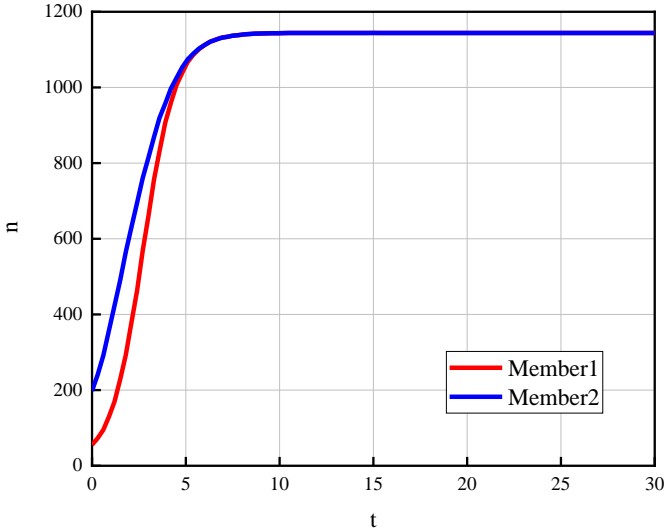

**Figure 7.** Symmetric mutualism mode (different initial knowledge amount).

### 5.4. Impact of Correlation Coefficient on Evolution of Symbiotic Network

According to the symbiotic coefficients: $\delta_{12} = (1 - \alpha_2) \times \beta \times \gamma_1 + \eta_2$, $\delta_{21} = (1 - \alpha_1) \times \beta \times \gamma_2 + \eta_1$, the symbiotic coefficients are the combined effects of provider's knowledge-based psychological personal ownership (KPPO), media richness, receiver's trust and organizational rewards matching.

### 5.4.1. Impact of Provider's Knowledge-Based Psychological Personal Ownership Coefficient

Assuming member 2 is the provider, taking $\alpha_1 = 0.5$, $\alpha_{21} = 0.2$, $\alpha_{22} = 0.5$, $\alpha_{23} = 0.9$; $\beta = 0.8$; $\gamma_1 = \gamma_2 = 0.9$; $\eta_1 = \eta_2 = 0$, the symbiotic evolution path of members 1 and 2 are simulated, and the results are shown in Figure 8. It can be seen from Figure 8 that when the KPPO coefficients of member 2 are 0.2, 0.5, and 0.9 (i.e., $\alpha_{21} = 0.2$, $\alpha_{22} = 0.5$ and $\alpha_{23} = 0.9$), the final stable knowledge levels of members 1 and 2 decrease gradually, and the change of member 1 is greater than that of member 2. It means that the KPPO of member 2 inhibits the knowledge growth of member 1; due to the reduction of organizational knowledge, the knowledge growth of member 2 also slows down. Thus, it is clear that the provider's KPPO can slow down knowledge growth and inhibit tacit knowledge transfer. The reason is that strong KPPO leads the provider to regard tacit knowledge as private property, resulting in knowledge hiding behavior and affecting the flow of knowledge within the organization. The tendency inspires organizations to establish corresponding incentive systems, encourage members to share knowledge actively, and then expand the knowledge amount to promote a steady improvement in organizational performance.

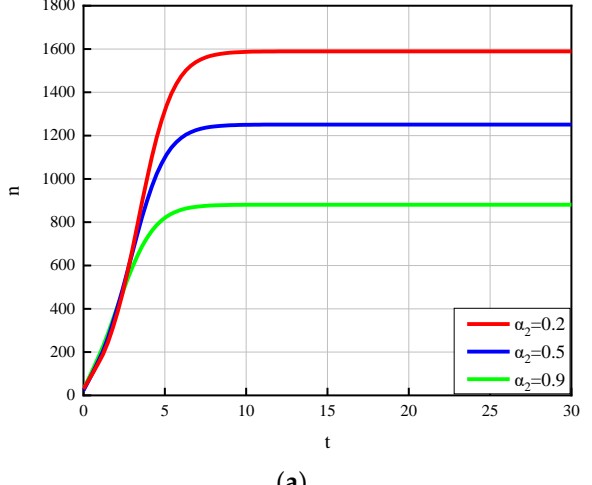 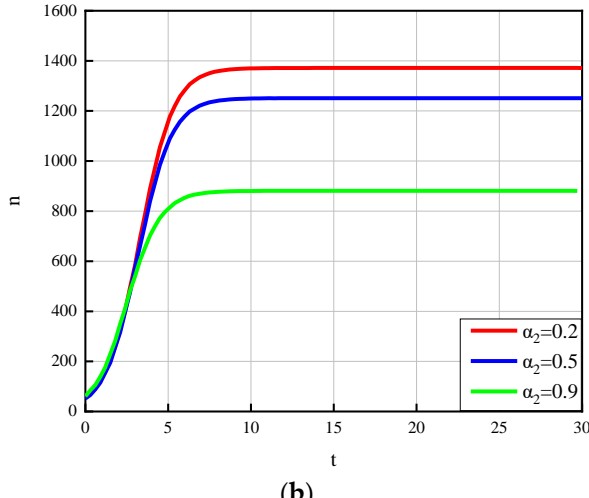

(**a**) (**b**)

**Figure 8.** Impact of member 2's KPPO ($\alpha_2$): (**a**) Impact of member 2's KPPO on member 1; (**b**) Impact of member 2's KPPO on member 2.

### 5.4.2. Impact of Media Richness Coefficient

Taking $\alpha_1 = \alpha_2 = 0.8$; $\beta_1 = 0.3$, $\beta_2 = 0.5$, $\beta_3 = 1$; $\gamma_1 = \gamma_2 = 0.9$; $\eta_1 = \eta_2 = 0$. The above analysis found that media richness causes knowledge reduction then affects the receiver's absorption. Assuming that member 1 is the receiver, member 1's knowledge evolution is shown in Figure 9. It can be seen from Figure 9 that when the media richness coefficients are 0.3, 0.5, 1 (i.e., $\beta_1 = 0.3$, $\beta_2 = 0.5$, $\beta_3 = 1$), the final stable knowledge levels of member 1 increase gradually, the larger the richness coefficient, and the larger the knowledge level. When the coefficient equals 1, knowledge is restored to 100%, and there is no loss. The receiver's absorption reaches the maximum. Therefore, this indicates that media richness can increase tacit knowledge amount. The reason is that high media richness reduces knowledge ambiguity, makes knowledge transfer activities more convenient, and promotes information transfer between knowledge subjects. This suggests that organizations should enrich transfer media and expand dissemination channels to avoid knowledge loss and promote efficient communication among members.

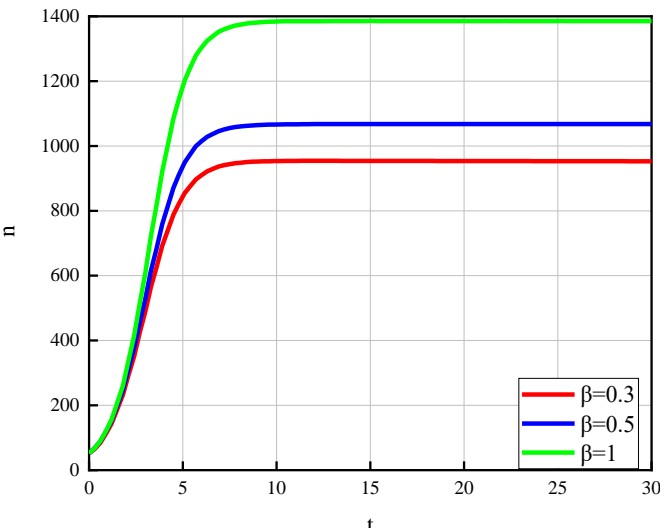

**Figure 9.** Impact of media richness ($\beta$) on member 1.

### 5.4.3. Impact of Receiver's Trust Coefficient

Taking $\alpha_1 = \alpha_2 = 0.5$; $\beta = 0.8$; $\gamma_{11} = 0.3$, $\gamma_{12} = 0.5$, $\gamma_{13} = 1$, $\gamma_2 = 0.9$; $\eta_1 = \eta_2 = 0$. The analysis above found that the receiver's trust affects their knowledge absorption. Assuming

that member 1 is a knowledge receiver, member 1's knowledge evolution is shown in Figure 10. It can be seen from Figure 10 that when the trust coefficients of member 1 are 0.3, 0.5, and 1 (i.e., $\gamma_{11} = 0.3$, $\gamma_{12} = 0.5$ and $\gamma_{13} = 1$), the final stable knowledge levels increase gradually, the larger the trust coefficient, the larger the knowledge level, when $\gamma_1 = 1$, member 1's absorbed amount reaches the maximum. Therefore, the results suggest that the receiver's trust can increase its tacit knowledge. The reasons are: firstly, receivers will consider the provided knowledge is valuable and accept it when they completely trust providers. Secondly, receivers' trust in providers also increases providers' willingness to share and facilitates knowledge transfer activities [67]. This tendency suggests that organizations should pay much attention to the symbiotic relationship among members, create a cooperative atmosphere, and improve trust among members.

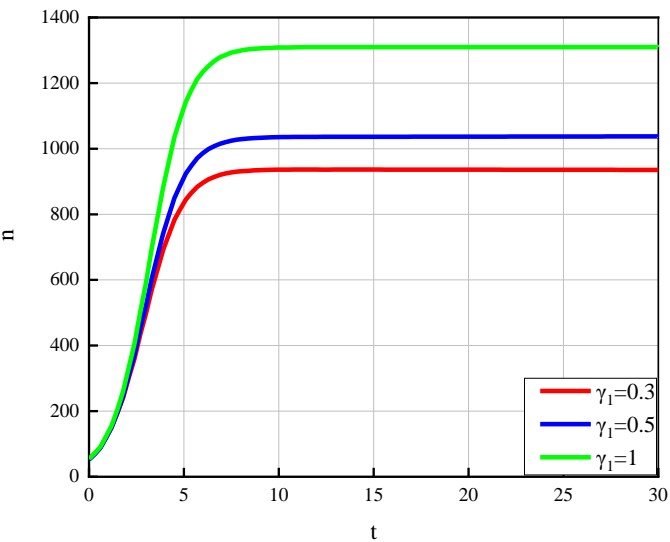

**Figure 10.** Impact of member 1's trust coefficient ($\gamma_1$) on member 1.

5.4.4. Impact of Organizational Rewards Matching Coefficient

Taking $\alpha_1 = \alpha_2 = 0.5$; $\beta = 0.8$; $\gamma_1 = \gamma_2 = 0.9$; $\eta_{11} = 0$, $\eta_{12} = 0.2$, $\eta_{13} = 0.5$, $\eta_2 = 0$. The co-evolution of members 1 and 2 are simulated, and the results are shown in Figure 11. It can be seen from Figure 11 that when the matching coefficients of member 1 are 0, 0.2, and 0.5 ($\eta_{11} = 0$, $\eta_{12} = 0.2$ and $\eta_{13} = 0.5$), the final stable knowledge levels of members 1 and 2 increase gradually, the larger the matching coefficient, the higher the knowledge levels. Therefore, this proves that organizational rewards matching can effectively increase tacit knowledge. The reason is that organizational rewards are the support and encouragement for knowledge transfer behavior. When organizational rewards match providers' needs highly, they will transfer their knowledge actively, while providers' knowledge increases with the whole network knowledge.

In reality, most organizations believe that tacit knowledge transfer is not an in-role behavior of members and usually do not take measures to stimulate the willingness of members to transfer knowledge [72]. However, this study finds that organizational rewards can promote tacit knowledge transfer behavior and increase organizational knowledge amount. Nevertheless, more attention should be paid to how well organizational rewards match individual needs. This encourages organizations to establish incentive systems to support knowledge transfer, including understanding members' needs and expanding reward types, which can improve the degree of matching.

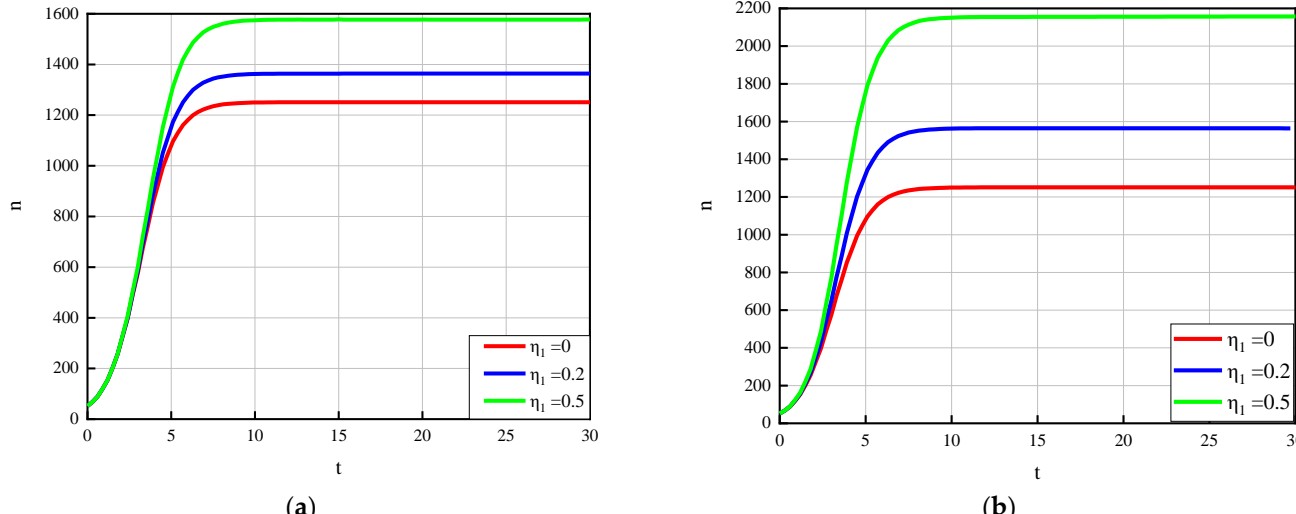

**Figure 11.** Impact of member 1's matching coefficient ($\eta_1$): (**a**) Impact of member 1's matching coefficient on member 1; (**b**) Impact of member 1's matching coefficient on member 2.

## 6. Conclusions

### 6.1. Conclusions

The transfer of tacit knowledge will lead to a higher growth rate of knowledge, and is the primary source of organizational innovation [1]. The organization's tacit knowledge transfer network (OTKTN) is a dynamic knowledge ecosystem composed of multiple knowledge subjects who transfer heterogeneous tacit knowledge to cope with a complex, competitive environment. The symbiotic relationships between knowledge subjects affect tacit knowledge transfer [32,33]. To clarify the mechanisms influencing symbiotic relationships on tacit knowledge transfer in the network, help members collaborate, and facilitate organizations to obtain sustainable competitive advantages, deep analysis of the symbiotic relationships among members is urgently needed. Firstly, the symbiotic perspective is introduced into OTKTN to constructing a network symbiotic system. Secondly, this article researches the process of knowledge transfer based on prior studies' views, variables such as knowledge-based psychological personal ownership (KPPO), media richness, trust, and organizational rewards matching; these are analyzed to construct symbiotic coefficients and to discuss symbiotic modes. Finally, numerical simulation is used to draw the evolution law of knowledge in detail. In summarizing the results, the following conclusions can be presented:

(1) Four levels have different influences on the evolution of tacit knowledge in the transfer process. By summarizing previous research conclusions, this article constructed a process model of tacit knowledge transfer within the symbiotic network and discussed the influence factors from four aspects: knowledge provider, knowledge receiver, media, and organization. This study shows that media richness, the receiver's trust, and organizational rewards matching all contribute effectively to increased members' tacit knowledge, but the provider's KPPO inhibits the increase of members' tacit knowledge. These are consistent with the conclusions of Wu et al. [18], Holste and Fields [23], Daft and Lengel [28] and extend the conclusion of Gagne [30]. The results show that a good atmosphere should be created to increase members' trust and collaboration. An organizational reward mechanism should be built to meet different needs and stimulate sharing behavior. Furthermore, the media should ideally be enriched to reduce knowledge transfer loss.

(2) Symmetric mutualism mode is the best mode between members. The result is consistent with the findings of previous studies. For example, Di and Dong [44], Ou et al. [46], and Yao and Zhou [47] opined that the mutualism mode is the best symbiotic mode. Through the analysis and discussion of all combinations of sym-

biotic coefficients, four symbiotic modes are determined. After the simulation of these modes, this study found that when in symmetric mutualism mode, knowledge subjects depend on each other and progress together. Their knowledge increments are equal. Organizational knowledge also increased because of members' knowledge transfer. The symbiotic relationship among members is more stable, which is conducive to the organization's sustainable development. The result shows that tacit knowledge transfer enjoys the influence of symbiotic relationships between members. Good relationships between them can significantly improve the efficiency of knowledge transfer. Thus, organizations should cultivate healthy and benign symbiotic relationships, promote knowledge subjects' positive behaviors, and shape good behavior norms. Doing so helps to accelerate knowledge flow and sharing.

(3)  In the symmetric mutualism mode, the evolution of tacit knowledge is affected by three factors. In order to analyze the influence of individual-related factors on tacit knowledge transfer under symmetric mutualism mode, this article discusses the influence of the maximum level in independence, initial knowledge amount, and natural growth rate on tacit knowledge transfer. The maximum level represents knowledge self-learning ability, initial knowledge represents knowledge stock, and natural growth rate represents knowledge absorption capacity. This study shows that the maximum level in independence mode positively impacts the final stable knowledge level, and the initial knowledge amount and natural growth rate positively impact the growth rate of knowledge, respectively. The results indicate that the higher the self-learning ability, the higher the knowledge increment, the larger the knowledge stock or stronger absorption ability, the faster the knowledge growth rate.

### *6.2. Implications*

The sustainable development of an organization is inseparable from innovation. Tacit knowledge is the crucial source of innovation, and knowledge transfer is the primary way to achieve the rapid growth of tacit knowledge. However, complex symbiotic relationships among knowledge subjects vastly affect the tacit knowledge transfer effect. Previous studies on tacit knowledge transfer only focus on the characteristics of the subject or object, fewer consider the role of symbiotic relationships between knowledge subjects. This article analyzed the influence mechanism of symbiotic relationships on the tacit knowledge transfer path in an organization's tacit knowledge transfer network. It provided suggestions for organizations to form a healthy and stable member relationship, expand organizational knowledge, and promote organizational innovation and sustainable development. This article enriches the related research on tacit knowledge transfer and has rich theoretical and practical implications.

### 6.2.1. Theoretical Implications

Tacit knowledge is a hot topic in knowledge management research. The conclusion of this article contributes to the existing studies from three aspects.

Firstly, based on the symbiotic perspective, this article analyzes the influence mechanism of symbiosis on tacit knowledge transfer. Available literature focuses on probing the influencing factors about tacit knowledge transfer from the single level, for instance, individual characteristics [18–25], media characteristics [26–28], and organizational characteristics [29–31]; there is a gap in researching on the role of symbiotic relationships among knowledge subjects. Knowledge transfer activity is a knowledge interaction among multiple knowledge subjects. Relationships between them will affect the effect of transfer. Therefore, this article elucidates the influence mechanism of members' symbiotic modes on the evolution of tacit knowledge and expands the study on tacit knowledge transfer.

Secondly, the symbiotic coefficient is analyzed. Previous studies on symbiosis only focused on the positive and negative symbiotic coefficients but did not further disassemble. This article constructs the symbiotic coefficient through in-depth analysis of the tacit

knowledge transfer process from four dimensions: knowledge providers' KPPO, media richness, receivers' trust, and organizational rewards matching.

Thirdly, extending the application of the Lotka–Volterra (L–V) model to the field of tacit knowledge management and solve the problem of complex data acquisition. Prior studies applied the L–V model to innovation ecosystems [44–47] or economies [48,49]. This article used the L–V model to investigate the symbiotic modes between knowledge subjects in the knowledge transfer network, which provides a new perspective for the subsequent research on this topic. Moreover, this article uses numerical simulation to simulate evolution results to solve the problem of difficult data acquisition.

6.2.2. Practical Implications

This article systematically and comprehensively discusses the symbiotic modes of organizational members and the impact on tacit knowledge transfer, providing a valuable reference for existing organizations to increase tacit knowledge.

Firstly, establish and improve incentive and evaluation mechanisms to mobilize members' sharing initiatives. Tacit knowledge is a scarce resource, but knowledge transfer needs costs. If there is no return for knowledge transfer, individuals are more inclined to hide knowledge. Therefore, organizations should recognize the importance of incentives for members, pay more attention to the match between incentives and individual needs, meet low-level needs (e.g., bonuses, wage increases, etc.) and high-level needs (e.g., enhancing job challenges, sense of professional achievement, giving some honors, etc.) respectively. However, organizational rewards need to depend on the contributed degree and implement differentiated rewards, which requires establishing a supporting evaluation mechanism, clear evaluation criteria, process, contributed degree, and reward class.

Secondly, enrich media and reduce knowledge loss. Daft et al. [27] found that fuzzy communication needs rich media. High-richness media help to reduce the ambiguity of information. This study found that the higher the media richness, the greater the tacit knowledge absorption. High-richness media reduces knowledge loss and makes knowledge transfer more efficient. Therefore, organizations should expand knowledge transfer channels and enrich communication media, for example, building knowledge bases or using advanced information technology to restore tacit knowledge.

Thirdly, create a harmonious symbiotic atmosphere and promote the symbiotic evolution of members. This study finds that symmetric mutualism is the best mode between members. Organizations should build a harmonious working environment and create a good atmosphere of mutual trust, concern, and support to make members synergistic progress. Organizations also need to guide members to share individual knowledge and experience, then integrate and preserve them as collective knowledge resources so that members' perceptions and opinions can be effectively applied by others [75].

Lastly, increase learning and training opportunities and improve members' tacit knowledge. This is necessary to provide members with systematic and appropriate training to improve their professional knowledge in an all-around way. Especially in today's era of the knowledge economy, competition is fierce, knowledge updates quickly, and continuous learning and strengthening training are also robust guarantees for self-improvement and organizational sustainability.

*6.3. Limitations and Further Research*

Limitations of this study lie in the following aspects. First of all, as a recessive resource, many factors affect tacit knowledge transfer. This article only elaborates from the four levels (knowledge provider, media, knowledge receiver, and organizational rewards); however, other factors also affect the tacit knowledge, such as managers' leadership and organizational culture. Further research can conduct an in-depth analysis of these other levels. Secondly, this article only discriminates the symbiotic relationships between the two knowledge subjects. The analysis of a multi-subject symbiotic relationship is of more practical significance. Therefore, further research should add new research subjects and

use the Lotka–Volterra model to explore the evolution process of multi-subject symbiotic relationships in a system.

**Author Contributions:** Conceptualization, J.X. and H.W.; methodology, J.X.; software, H.W.; writing—original draft preparation, H.W.; writing—review and editing, J.Z.; visualization, H.W.; supervision, J.X.; project administration, J.Z.; funding acquisition, J.Z. All authors have read and agreed to the published version of the manuscript.

**Funding:** The research was supported by the National Social Science Fund of China with grant number 19BTQ035.

**Institutional Review Board Statement:** Not applicable.

**Informed Consent Statement:** Not applicable.

**Data Availability Statement:** The data presented in this study are available in this paper.

**Conflicts of Interest:** The authors declare no conflict of interest.

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
