# Peer review of "Innovation Research on Symbiotic Relationship of Organization’s Tacit Knowledge Transfer Network"

_sustainability, doi:10.3390/su14053094_

Round 1

Reviewer 1 Report

Dear authors,

Your manuscript is well-written and discusses on an interesting topic of tacit knowledge. However, a minor revision as per the file attached is required to further improved the manuscript.

Best wishes!

Reviewer 2 Report

Dear authors,

This manuscript deals with relevant issues, and it presents scientific soundness.

However, I recommend mainly the following improvements:

  • Clarify the problems linked to the topic before to present the aim and objectives.
  • For a better understanding, suggest differentiate the discussion and conclusions sections.

Reviewer 3 Report

  • Line 163: Vito Volterra, not Victor Volteler (https://www.britannica.com/biography/Vito-Volterra)
  • ines 219 - 222: the authors write about saturation, but do not take it into account in measuring transfer of knowledge. For instance using their model, if the flow of knowledge is perfect (or near-perfect), which corresponds to delta coefficients equal to 1, the knowledge of both members increases to infinity at an exponential rate.
  • Page 8 – double equation no. 3
  • pages 8-9: In the equations, the coefficient of influence of participant 1 on participant 2 does not depend in any way on the relations between them. There is a parameter depending only on participant 1, two parameters depend only on participant 2 and one parameter is the same for all participants, but there is nothing that takes into account the communication between them, which seems like it does not reflect reality.
  • page 8, bottom equation: lack of Σ in the first line.
  • I would suggest having a clear summary of what is included in the relationship factors. It would be clearer for the readers if that information was contained in one place and not on the length of three pages.

Reviewer 4 Report

Dear Authors,

I have read your article and I appreciate your extensive work. At the present, I think that one crucial suggestion whould be useful to address the text:

The whole text is unbalanced on the analysis of relational behaviors and less on arguing WHY a symbiotic relationship can affect the sustainability of organizations, moreover, the logical construction is weak, because it rests on a single reference published in the same journal (Zhang XX, Gao CY, Zhang SC Research on the knowledge sharing incentive of the cross-boundary alliance symbiotic system. Sustainability 2021, 13, 10432).
